# Association between Cervical Microbiota and HPV: Could This Be the Key to Complete Cervical Cancer Eradication?

**DOI:** 10.3390/biology11081114

**Published:** 2022-07-26

**Authors:** Eliano Cascardi, Gerardo Cazzato, Antonella Daniele, Erica Silvestris, Gennaro Cormio, Giovanni Di Vagno, Antonio Malvasi, Vera Loizzi, Salvatore Scacco, Vincenzo Pinto, Ettore Cicinelli, Eugenio Maiorano, Giuseppe Ingravallo, Leonardo Resta, Carla Minoia, Miriam Dellino

**Affiliations:** 1Department of Medical Sciences, University of Turin, 10124 Turin, Italy; 2Pathology Unit, FPO-IRCCS Candiolo Cancer Institute, Str. Provinciale 142 km 3.95, 10060 Candiolo, Italy; 3Department of Biomedical Sciences and Human Oncology, Obstetrics and Gynecology Section, University of Bari “Aldo Moro”, Piazza Aldo Moro, 70100 Bari, Italy; g.cazzato3@studenti.uniba.it (G.C.); antoniomalvasi@gmail.com (A.M.); vincenzo.pinto@uniba.it (V.P.); ettore.cicinelli@uniba.it (E.C.); miriamdellino@hotmail.it (M.D.); 4Department of Emergency and Organ Transplantation, Pathology Section, University of Bari “Aldo Moro”, Piazza Giulio Cesare 11, 70124 Bari, Italy; eugenio.maiorano@uniba.it (E.M.); giuseppe.ingravallo@uniba.it (G.I.); leonardo.resta@uniba.it (L.R.); 5Experimental Oncology, Center for Study of Heredo-Familial Tumors, IRCCS Istituto Tumori Giovanni Paolo II, 70124 Bari, Italy; anto.dani27@gmail.com; 6Gynecologic Oncology Unit, IRCCS Istituto Tumori Giovanni Paolo II, 70124 Bari, Italy; ericasilvestris85@gmail.com; 7Gynecologic Oncology Unit, IRCCS Istituto Tumori Giovanni Paolo II, Department of Interdisciplinary Medicine (DIM), University of Bari “Aldo Moro”, 70124 Bari, Italy; gennaro.cormio@uniba.it (G.C.); vera.loizzi@uniba.it (V.L.); 8Clinic of Obstetrics and Gynecology, “San Paolo” Hospital, 70124 Bari, Italy; giovanni.divagno@asl.bari.it; 9Department of Basic Medical Sciences and Neurosciences, University of Bari “Aldo Moro”, Piazza Giulio Cesare 11, 70124 Bari, Italy; salvatore.scacco@uniba.it; 10Unit of Hematology and Cell Therapy, Laboratory of Hematological Diagnostics and Cell Characterization, 70124 Bari, Italy; carlaminoia@libero.it

**Keywords:** microbiota, cervical microbiota, cervical intraepithelial neoplasia, CIN, cervical cancer

## Abstract

**Simple Summary:**

The microbiota can modulate immune responses and modify the physiology of the human organism, thereby increasing infective risks and a neoplastic predisposition. In this review, we focus on the composition of the cervical microbiota, to identify the risk of developing Cervical Intraepithelial Neoplasia and better understand the interaction between cervico-vaginal microbiota and human papillomavirus as a means of promoting the identification of new therapeutic strategies. In fact, no therapy for HPV is yet available. A better understanding of the cervical micro-environment could be a key element allowing complete viral clearance to be achieved in largely affected populations.

**Abstract:**

The heterogeneity of the cervico-vaginal microbiota can be appreciated in various conditions, both pathological and non-pathological, and can vary according to biological and environmental factors. Attempts are still in course to define the interaction and role of the various factors that constitute this community of commensals in immune protection, inflammatory processes, and the onset of precancerous lesions of the cervical epithelium. Despite the many studies on the relationship between microbiota, immunity, and HPV-related cervical tumors, further aspects still need to be probed. In this review article, we will examine the principal characteristics of microorganisms commonly found in cervico-vaginal specimens (i) the factors that notoriously condition the diversity and composition of microbiota, (ii) the role that some families of organisms may play in the onset of HPV-dysplastic lesions and in neoplastic progression, and (iii) possible diagnostic-therapeutic approaches.

## 1. Epidemiology of HPV

Papillomaviruses are widely distributed in mammals and are species-specific [1]. Human papillomavirus (HPV) cannot be cultivated in tissue cultures or in common experimental animals [1]. They are members of the Papillomaviridae family, have no coating, measure from 50 to 55 nm in diameter, and have an icosahedral capsid of 72 capsomers [2]. Of the approximately 200 genotypes of HPV, subdivided into 14 species, about 40 can infect the epithelial cells (skin or mucous membranes) of the anogenital regions and other areas [3]. Differentiation into types is made based on the characteristics of the L1 protein [4]. HPV was the first virus to be recognized as responsible for cervical cancer (CC). According to the degree of association with invasive tumors, HPV genotypes have been subdivided into: high oncogenic risk (16, 18, 31, 33, 35, 39, 45, 51, 52, 56, 58, 59, 66, 68), related with an increased risk of developing CC [5]; low oncogenic risk (6, 11, 40, 42, 43, 44, 54, 61, 70, 72, 81, 89) associated with no disease most commonly or with benign epithelial lesions (such as anogenital and oropharyngeal warts) [6]; and, finally, HPV with an “undetermined risk” (3, 7, 10, 27, 28, 29, 30, 32, 34, 55, 57, 62, 67, 69, 71, 74, 77, 83, 84, 85, 86, 87, 90, 91) include those whose oncogenicity has not yet been fully defined [7]. It has been established that about 99.8% of CC have a high-risk HPV DeoxyriboNucleic Acid (DNA) sequence, particularly HPV 16 and 18, found in about 70% of invasive carcinomas [8]. The prevalence of infection is very high (70% of sexually active female patients over 25 years old), and most infections tend to regress spontaneously, with or without manifestations of dysplasia; only in some cases can HPV infection become persistent [9]. Data from the scientific literature show that in female patients over 30 years old, persistent high-risk HPV infections play a critical role in predicting the risk of developing CC [8,9,10]. The risk of developing in histopathologic high-grade cervical intraepithelial lesion HSIL (CIN2 and CIN3) or invasive CC is estimated to be much higher in women with persistent high-risk HPV infection, being 11 times higher in the 30–44 age group, 35-fold higher between 45 and 54 years, and 49-fold in those over 50 years of age [11]. The ability of HPV viruses, especially those at high risk, to integrate into infected cells, and to orchestrate a gene expression program that allows the transcription of oncogenic proteins (E6, E7), promotes carcinogenicity [12]. Cervical intraepithelial lesions (CIN) can regress spontaneously, or progress to invasive neoplasia in different percentages depending on their severity. More specifically, histopathologic low-grade cervical intraepithelial lesion LSIL (CIN1) tends to regress spontaneously, particularly in young patients. Ostor and coworkers [13] report that CIN1 subsides spontaneously in 60% of cases, persists in 30%, and can progress in 10% of cases. On the other hand, CIN2 regresses spontaneously in 40%, persists in another 40% of cases, and can progress in 20% of cases; finally, CIN3 can regress in 33% of cases and progress in more than 12% of cases.

## 2. Screening and Histopathology of Cervical HPV Lesions

The natural history of CC is typically characterized by the progression, over the years, of non-invasive HPV-related precancerous lesions to invasive carcinoma [7,8,9,10,11,12,13]. Cytology (Pap-Test) and human papillomavirus detection (HPV-DNA) are two screening tests whose purpose is to detect CC or precancerous changes at an early stage. Before the development of the HPV-DNA test, the Pap-Test alone was performed every 3 years in women after the onset of sexual activity or in any case from 25 years of age [14]. Today, according to guidelines from different countries, HPV-DNA tests are used for women over 30 or over 25, and a Pap-Test is only done if it gives a positive result [15,16,17,18,19,20]. In fact, globally the recommended method of primary CC screening is the HPV-DNA test, independently of resource settings [21], due to its sensitivity compared to the Pap-Test, even in the presence of a lower specificity especially for the identification of CIN2 and CIN3 lesions. HPV testing is recommended by the World Health Organization [14] and other guidelines [15,22] even with respect to the Pap-Test which is now considered a secondary test [14]. In fact, the Pap-Test even if it has allowed to tangibly reduce the incidence of invasive carcinoma, however, presenting a variability between different operators, it can lead to the diagnosis of false negatives as well as cases of invasive CC are also reported in the literature in women regularly investigated with the Pap-Test [23]. In view of the crucial role of persistent infection of hr-HPV, the focus has shifted to the use of HPV-DNA testing as a screening test [24] so that access to treatment is increasingly targeted and timely. According to the ASCO guidelines, if the HPV-DNA test results positive, genotyping for HPV 16/18 (with or without HPV 45) and/or Pap-Test are also indicated [21]. In the event of a positive or abnormal result, the HPV-DNA test procedure involves colposcopy and related biopsy [21]. Conversely, in discordant results between the HPV test and cytological examination, it is recommended to repeat the HPV-DNA test one year later, then repeat the test at 12–24 months in case of negativity or colposcopy in women who tested positive [21]. Finally, in cases of CIN2 histological diagnosis, patients should be offered a surgical solution followed by targeted follow-up over time [21]. According to data from the randomized study published by Ronco et al. [25], HPV-research-based screening is more effective than the Pap-Test in preventing CC in women aged 25–60, because it allows an earlier identification of high-grade persistent lesions. In fact, the execution of the HPV-DNA test is useful in stratifying the population according to the degree of risk: a negative test indicates a low risk of developing CC, and so in these controls can be made at longer intervals. Although with differences between the various settings, the HPV-DNA test should be started from the age of 30 in the general female population, regular screening being done with the HPV test validated every 5–10 years, versus 25 years of age in women living with HIV, who should be screened more frequently, every 3–5 years. CC is the fourth most common malignancy among women worldwide, accounting for approximately 7% of all female cancers [26,27]. As reported in the literature, most of the diagnoses of CC can be associated with the presence of HPV infection and, in some studies, these associations can reach levels comparable to almost all cases [28]. Among the high-risk HPV genotypes, variant 16 has the highest affinity for neoplastic progression with over 50% of cases, followed by variant 18 which occurs in 20% of cases; this association tends to vanish in the remaining high-risk genotype up to 5% and even less [29]. Conversely, there does not appear to be a significant difference between the HPV status and the histotype of the carcinomas except for squamous-cell carcinoma (SCC) which is unlikely to be HPV-negative [30], as well as in mixed adeno-squamous form where the HPV positivity may reach up to 86% [31] of cases and also in the vast majority of Adenocarcinoma in situ (AIS) [30]. Vice versa, the prevalence of HPV among adenocarcinoma (AD) types can vary and, according to the International Endocervical Adenocarcinoma Criteria and Classification, ADs are divided in two categories: HPV-associated (HPVA) and not HPV-associated (NHPVA), with well-defined characteristics due to histology HPVA shows more mitotic activity or apoptotic figures than NHPVA. If focal or equivocal HPVA features are visible at ×200, a tumor can be classified as a “limited HPVA” and provisionally diagnosed as NHPVA AD [32]. To this histotype belong different variants such as mucinous (that can show aspects of HPVA and NHPVA), gastric (prevalent NHPVA type), endometrioid and serous carcinomas (extraordinarily rare). NHPVA comprehends histological variants [32] such as gastric, clear cell, serous, endometrioid, or mesonephric carcinomas that notoriously tend to be HPV negative [33] compared to histotypes presenting glandular/villo-glandular/intestinal aspects that appear to have a higher percentage of HPV positivity. 

### Principal Biomarkers for Cervical Cancer

The study of viral and cellular biomarkers that could be useful for identification of specific stages of cervical intraepithelial lesions related to hrHPV infection is closely linked to the biology of HPV and the different stages from infection through intraepithelial lesion, and, if this is not properly treated, to invasive CC. After hrHPV infection, the infected cervical epithelium begins to proliferate, leading to the transformation into CIN1, 2, and 3 [34,35,36,37]. Progression from a transient to a transformation HPV infection is characterized by a sharp increase in HPV mRNA E6/E7 and protein expression [34,37]. With this in mind, several authors have assessed how the detection of mRNA transcripts belonging to E6/E7 proteins could be helpful for identifying cervical precancers [36,37,38,39,40,41,42]. From these works, we can deduce a good sensitivity and specificity of these tests; there are two platforms used for the study of this biomarker (PreTect. Proofer and APTIMA GenProbe). The p16INK4a protein is a cyclin-dependent kinase inhibitor that plays a key role in cell cycle regulation and is upregulated when E7 is overexpressed as in HPV infections, representing an ideal biomarker to define the nature of cervical lesions [43,44], especially if combined in a single test together with Ki-67 [45,46] which is generally used in immunohistochemistry to evaluate the cell proliferation index [47]. Double-stained cytology p16/Ki-67 (approved by FDA on 3 October 2020 [48]) is a qualitative immunocytochemical assay intended for the simultaneous detection of proteins P16INK4a (clone E6H4) and Ki-67 (clone 274-11AC3V1) in cervical specimens of women aged 25–65 years with positive HPV test (high oncogenic risk); it is also indicated for patients aged between 30 and 65 who have to postpone colposcopy or have other risk factors regardless of the result of the HPV test [48]. It is highly expressed in almost 100% of cases of CIN2, CIN3, squamous CC, but is rarely found in benign forms; it is highly expressed in 100% of AIS cases. Several studies have highlighted the ability of the p16 test to identify the neoplastic transformation of cervical cells infected with human papillomavirus, being able to predict with greater precision the underlying cervical intraepithelial neoplasia of grade 3 or worse [49,50]. Specifically, the ATHENA study, on a group of 7727 patients, showed that p16/Ki-67 dual-stained cytology was significantly (*p* < 0.0001) more sensitive than Pap-Test (74.9% vs. 51.9%) for the triage of HPV-positive women and that specificity was comparable between the two methods [50]. In studies directed by Bergeron and Petry [46,51], as well as by Wright and coworkers, the authors described a p16/Ki-67 dual-stained cytology, that either alone or combined with HPV16/18 genotyping, represents a promising approach as a sensitive and efficient triage for colposcopy of HPV-positive women when primary HPV screening is utilized [50]. These studies led the way in using p16/ki-67 as a reliable tool for risk stratification of HPV-positive women with cervical lesions, reporting elevated sensitivity values similar to those of the HPV test [51,52,53,54,55]. Furthermore, other authors have shown that p16/ki-67 had a higher specificity than the HPV test [45,53,54,55]. These data reinforce the diagnostic value of p16/Ki-67, which represents a useful tool in avoiding further diagnostic investigations such as unnecessary colposcopies given its ability to more precisely identify female patients at increased CIN2 risk. In the field of research into serum biomarkers for the early detection of CC, the activation of Macrophage-Colony Stimulating Factor (M-CSF) and Vascular Endothelial Growth Factor (VEGF) is likely involved in the pathogenesis and spread of CC. In particular, M-CSF is overexpressed in the CC lines compared to CIN and the blocking of the M-CSF receptor determines both a growth arrest and an increase in intratumoral apoptosis [56,57]. In various papers, Lawicki and coworkers [38,58], using the immunosorbent assay (ELAISA), evaluated the plasma levels of M-CSF as compared to commonly accepted tumor markers, such as CA 125 and SCC-Ag in CC patients before surgery and in healthy subjects. The M-CSF plasma level was significantly higher, as also CA 125 and SCC-Ag, in CC patients. The diagnostic sensitivity of M-CSF was higher than stem cell factor, CA 125, and SCC-Ag (25%, 30%, 40%, respectively). The diagnostic specificity was high and equal for all tested cytokines and CA 125 (92%). Positive and negative predictive values were higher for all tested parameters but highest for M-CSF (83% and 69.7%, respectively) [38]. More recent studies [59,60,61,62] confirmed that cytokine members of hematopoietic growth factors may have a diagnostic potential in CC. Zajkowska evaluated serological markers against CA 125 and SCC-Ag in 100 CC patients with chemiluminescent microparticle immunoassay and showed statistically significant M-CSF values from all parameters tested in the CC cohort compared to the control groups [59]. Similar results have also been described by Lubowicka et al. who investigated M-CSF, matrix metalloproteinase-2, and its inhibitor beyond CA 125 and SCC-Ag in 89 CC patients and in 50 healthy women (aged 22–61 years) reporting for M-CSF the highest specificity (86%) in the CC group [60]. Moreover, the association of these different markers increased in specificity, as observed in the combination between matrix metalloproteinase-2 and CA 125 in different CC stages [60]. While median levels of M-CSF and VEGF, as well as CA 125 and SCC-Ag are shown to be significantly different in women with CC, this relationship does not seem to be specific to SCC alone; in fact, the plasma levels of M-CSF and VEGF are also higher in AD than in the control group and, moreover, no significant differences were observed between SCC and AD [61]. Sidorkiewicz reported 81% sensitivity and 74% specificity for VEGF in the SCC group and 86% and 76% in the AD group [61]. Although the results of these studies are encouraging and suggest a diagnostic utility and a possible clinical applicability of serum biomarkers in patients with CC, to date they are not yet sufficiently studied, and more confirmations would be useful for their wider use than p16/Ki67 testing and mRNA detection.

## 3. Diversity and Influencing Factors of Vaginal Microbiota

The vaginal microbiota (VM) is characterized by a heterogeneous variety of microorganisms that are commonly found in cervicovaginal samples of female patients in both pathological and non-pathological conditions. Based on species composition and the relative abundance of bacterial population present in genital mucosae of 396 healthy women [40], using next-generation sequencing platforms, Ravel et al. analyzed and classified the microbiota in five clusters, also known as community state types (CSTs) [41]. In CST I, II, III, and V, *Lactobacillus (L.) crispatus*, *L. gasseri*, *L. iners*, and *L. jensenii* are more common, respectively, unlike CST IV, which features a high bacterial diversity including several anaerobic species such as *Gardnerella*, *Megasphera*, *Atopobium*, *Prevotella*, *besides Pseudomonas*, *Shigella*, *Peptostreptococcus*, *Enterococcus*, *Streptococcus*, *Propionibacterium*, *Bifidobacterium*, and *Brevibacterium* species [42]. The attempt to classify vaginal microorganisms in order to establish the role of VM in immune protection, inflammatory processes, and in cancer genesis is still ongoing today, owing to the different factors that notoriously influence the diversity and composition of VM [63,64]. The diversity of vaginal microbiota has a great geographic and ethnic variability [65,66] showing a greater variety in African women. Again, *Lactobacillus* species are significantly more frequent in Caucasian and Asian populations compared to Hispanic areas [41]. These differences, consisting of high VM diversity and variable abundances of *Lactobacillus*, called dysbiosis, may be due to hygiene practices, social, metabolic, and immune factors that influence vaginal homeostasis causing bacterial vaginosis (BV), with or without symptoms [67,68,69]. In this regard *L. iners*, that is unable to produce D-lactic acid and H_2_O_2_, seems to favor dysbiosis [70,71], although generally species of *Lactobacillus* produce lactic acid, hydrogen peroxide, and antimicrobial peptides, such as bacteriocins and biosurfactants, that inhibit the growth of bacteria and viruses [42,71,72], and regulate vaginal homeostasis. Vice versa, *L. iners* produces L-lactic acid and inerolysin (a pore-forming toxin cytolysin capable, like *Gardnerella*-released vaginosis, of prejudicing the integrity of the vaginal epithelium); both of these promote pathogenic proliferations and infections [70,73,74]. Although hormonal endogenous agents have a rather small impact on cervical and vaginal cancers’ development, female hormones and reproductive factors can still induce diverse female malignancies [75,76,77] and are one of the most important endogenous agents in the composition of the VM. In fact, estrogen receptors are widely expressed in the endometrium, where they regulate the growth and proliferation of epithelial cells and condition the stability of VM; estrogen-related factors such as menstruation, sexual activity [78], oral contraceptive use, pregnancy and lactation, or diseases such as diabetes mellitus and stress, cause hormonal fluctuations that influence the biological presence and the abundance of a bacterial community [71]. Age is clearly a crucial factor in relation to the action of the mechanisms underlying endocrine stimulation: low estrogen levels after birth cause a reduction in vaginal *Lactobacillus* and an increase in anaerobic populations until puberty [42], when there is an increase in estrogen and glycogen, further processed in lactic acid by *Lactobacillus* species [71,79]. During the reproductive age, the cyclic secretion of female hormones determines a great instability, especially during the menstrual phase when both the low levels of estrogen and progesterone, and the presence of menstrual blood, are linked to the decrease in some microorganisms and the enrichment of others [79,80]. Instead, high levels of estrogen and progesterone are a favorable factor for the stability of the VM during the female fertile age [79]. On the other hand, as described by David A MacIntyre et al. [81], a 100–1000-fold decrease in circulating estrogen concentrations postpartum triggers a significant increase in VM diversity. This condition also occurs in the menopausal state, when the sharp decrease in the production of female hormones causes atrophy of the vaginal epithelium and a relative absence of glycogen and *Lactobacillus* species [82,83,84]. Furthermore, exogenous factors can also influence the composition of the vaginal microbiota: the use of oral contraceptives is associated with an increased level of inflammatory cytokines in the cervix [85]. In fact, it has been shown that the use of oral contraceptives is associated with the risk of CC [86], even apart from the role played by the inflammatory microenvironment in influencing cell proliferation in different models of solid tumors [77,87,88,89,90]. In a meta-analysis conducted on the use of both combined hormonal contraceptives and progesterone-only hormonal contraceptives, Lenka A Vodstrcil et al. showed that, in general, they are equivalent, reducing recurrent BV by 31 and 32%, respectively [91]. These results are also reinforced by other studies showing the propensity of oral contraceptives to favor *lactobacillus* species [91,92,93], unlike contraceptives such as the intrauterine device and medroxyprogesterone acetate, which do not appear to affect the composition of the VM [94]. Furthermore, as previously mentioned, other exogenous factors seem to influence the biodiversity of the VM: in fact, in smoker women there is a greater diversity of species characterizing the VM, with a substantial reduction in *L. crispatus* [95] and an increase in species belonging to CST IV; *Peptostreptococcus* and *Veillonella* have been identified as most closely related to the use of tobacco [95]. Finally, even vaginal douches can affect not only the composition of the VM [96,97,98] but also the risk of related HPV infections or HPV-associated cervical diseases [98,99]. In fact, literature reports described a high production of proinflammatory cytokines in women with CST IV, which increased the recruitment of activated CD4+CCR5+ cells to the vaginal mucosa, increasing the risk of damage to the epithelial barrier and promoting HPV infection [42].

## 4. The Association of Cervical Microbiota with the Risk of CIN

The human body is an extraordinary ecosystem constituted by trillions of microorganisms. The coevolution that occurred between man and microbes formed a complex communication network [100,101]. Most microorganisms are bacteria that live in the gut and have a strong impact on the health of their hosts [102]. Joshua Lederberg summed up this concept in 2001, when he coined the term “microbiota” referring to the community of commensal, symbiotic, and pathogenic microorganisms which share the same space and exert diversified interactions with the specific human tissue compartments where they are hosted [103]. In 1977, Carl R. Woese and George E. Fox had reported the first study utilizing the 16S rRNA gene to recognize these bacteria, and demonstrated that the gene could be used to detect microorganisms on the basis of molecular phylogeny [104]. This approach was innovative in the biology field because previously this gene had been largely used only for bacteria identification. In the literature, it has recently been shown that the microbiota can play a crucial role in cancer progression, by affecting the host immunity. The influence of the microbiota on HPV-induced gynecological neoplasms remains poorly understood. Therefore, several studies have focused on the VM since together with immune regulation, it could play an important role in HPV carcinogenesis [105]. The cervical microenvironment is complex, constituted by immune cells and the specific microbiota that regulate local immune responses [106]. Indeed, the immunologic status of the host and HPV-induced immune evasion could explain persistent HPV infection, but alone, these factors are not sufficient to determine cancer development [107]. In fact, not all HPV-infected patients develop CC since the immune system together with other additional factors influence the progression of cervical intraepithelial lesions to CC or else regression [108,109,110]. Therefore, several researchers explored the relationship between the VM and HPV infection, as the microbiota would seem to play a decisive role in neoplastic evolution [42,111,112]. Recently, studying virus–bacteria–host interaction models, two models were evaluated with the aim of better understanding the microbiota mechanisms in the development of virus-associated cancers [113]. The first model relied on the concept that the microbiota could affect viral infectivity through the release of bioproducts that could modulate virus–host interactions. The second model was based on the concept that bacteria–host interactions impact the host gene expression, increasing viral production and promoting tumorigenesis associated with the viral infection [114]. One of these studies characterized the microbiota in a population with HPV infection, using laboratory culture, and reported on the presence of ‘abnormal vaginal flora’ (that subsequent investigations would then identify using 16S rRNA analysis) [115,116]. Based on the microbiota composition, the microenvironment could also protect the host from viral infections. It is known that the main defense mechanisms of the cervicovaginal mucosa are antimicrobial peptides, microbiota dominated by lactobacilli and a pH of less than 4.5. An alteration of these protection mechanisms can result in physiochemical modifications that damage the cervical epithelium and vaginal mucosa [117]. Particularly, a reduction in lactic-acid-producing Lactobacilli, with a consequent increase in vaginal pH (4.5), can induce atypical bacterial growth and a decrease in protective flora [118], weakening the endogenous system of defense against viral infection. The most common type of cervicovaginal dysbiosis is BV, defined as a VM with scarce *Lactobacillus*, and an increase in anaerobes [119]. BV is associated with an increase in the levels of proinflammatory cytokines such as IL-1b and a decrease in the levels of the anti-inflammatory molecule, secretory leukocyte protease inhibitor, supporting the theory that BV causes changes in the immune system that can lead to a greater susceptibility to HPV and hence the development of high-grade cervical dysplasia [120]. Particularly, the authors also studied the oxidative DNA damage related to the natural history of HPV persistence and cervical carcinogenesis. BV-associated oxidative stress [121] could be involved in the generation of reactive oxygen species that could generate double-stranded DNA breaks in the host genome, as well as the HPV episome, accelerating HPV integration and neoplastic transformation, a mechanism also employed by the HPV E6 oncoprotein [122]. The integration produces a loss of E1 and E2 genes, which regulate E6 and E7 transcription. Subsequently, transcription of these oncoproteins goes unchecked after viral integration, leading to a greater cellular proliferation and a reduction in apoptosis [123]. Several studies in the literature support this evidence. Particularly, Mitra et al. assessed a group of 169 patients referred for colposcopy, who showed an increased bacterial diversity connected with diminished lactobacilli related to the gravity of the cytological lesion [124]. Oh et al. also highlighted the finding that the risk of L-SIL in HPV-positive patients with a high-risk microbial pattern was suggestively higher than in HPV-negative women with a low-risk microbial score (or: 34.1, 95% CI: 4.95–284.5). Therefore, the *Lactobacillus* genus (first described in 1892 by Döderlein) is usually abundant in the cervicovaginal microbiota [125]. *Lactobacillus* species such as *L. crispatus*, *L. gasseri*, and *L. jensenii* could generate lactic acid and hydrogen peroxide (H_2_O_2_), which limit the progression of viral and bacterial infections [71] (Figure 1).

Therefore, the microbiota is the first line of defense against infections. The second line is its composition, since it can produce lactic acid and H_2_O_2_, that have a defensive role against viral and bacterial infections [70]. On the contrary, *L. iners* was evaluated as a transitional species leading to the dysbiosis state [126]. Cross-sectional studies show a negative relationship between HPV infection and CIN with *Lactobacillus* dominance, except for *L. iners*, which shows the opposite tendency, being correlated with a higher frequency of HIV, HPV, and HSV-2 [126]. Therefore, in most HPV-infected women, the immune response can limit the infection, preventing high-grade lesions and tumors [127]. On the other hand, some dysbiotic bacterial communities seem to cause immune dysregulation, favoring a tumor-promoting microenvironment [128], and may play an important role in CC progression [129]. Indeed *L. iners* are frequently detected in patients diagnosed with CIN [130]. However, the role of this bacterium in cervicovaginal health is still uncertain, since it can be found in normal environments as well as in states of vaginal dysbiosis [131]. In fact, it has been demonstrated that *L. iners* has more complex nutritional needs and a more variable morphology than other *Lactobacillus* species. It has an unusually small genome, suggestive of a symbiotic or parasitic lifestyle [74]. Actually, *L. iners* may have clonal variants that in some cases promote health and in other cases are associated with dysbiosis and a predisposition to disease [74]. Moreover, literature reports suggest that the VM is different in female patients of diverse ethnicities and such variety may be associated to genetic differences among races including a few mitochondrial DNA haplotypes [131]. Consequently, distinguishing the VM in female patients of each race could justify the differing incidence of BV and sexually transmitted infections among diverse ethnicities and reveal the importance of genetic factors in influencing the VM of some individuals, making them prone to illnesses [130]. Female patients with dysbiosis can develop chronic inflammation, which may be an important factor for cancer development in different tissue types, including cervical tissue [42]. Moreover, in CST IV, there is also *Gardnerella vaginalis*, which produces sialidase that degrades the vaginal mucosal surface posing a physical barrier able to inhibit bacteria–host interactions [67,132]. Gao and coworkers tested 70 healthy women (32 HPV-negative and 38 HPV-positive) with normal cervical cytology and, using the Shannon–Weiner diversity index, they discovered that HPV-positive female patients had a greater biological diversity [91]. The major CSTs found in female patients with CIN were CSTs, characterized by *Lactobacillus* depletion, anaerobic bacteria predominance, and *L. iners* dominance [133]. On the other hand, *L. crispatus* was the predominant *Lactobacillus* species in Italian female patients, who were better able to clear HPV infection [133]. Indeed, Kwasniewski et al. [134] reported a study of the vaginal flora of 250 female patients, including 85 women with high-grade SIL and HPV positivity, 95 female patients with low-grade SIL and HPV positivity, and 70 healthy controls. In the control group, high levels of *L. crispatus*, *L. iners*, and *L. taiwanensis* and an absence of *Gardnerella vaginalis* and *L. acidophilus* were identified. In the low-grade-SIL group, *L. crispatus* was less numerous than in the control group and *L. acidophilus* and *L. iners* prevailed. In the high-grade-SIL group, *Gardnerella vaginalis* and *L. acidophilus* were prevalent, while the frequencies of *L. iners, L. crispatus,* and *L. taiwanensis* were lower than in the control group. Therefore, these results show a possible relationship between the VM, HPV infection, and CIN development. In particular, a microbiota dominated by *Gardnerella vaginalis* and with reduced *L. iners*, *L. crispatus*, and *L. taiwanensis* may play a role in HPV persistence, CIN development, and CC [134]. Additionally, Oh et al. [125] reported a cytology study on women with LSIL or HSIL vs. normal controls. The outcomes suggested that a paucity of *L. crispatus* and a predominance of A. vaginae and secondarily of *Gardnerella vaginalis*, and *L. iners* were linked to an almost 6-fold increase of the risk of cervical LSIL/HSIL disease (odds ratio (OR): 5.80, 95% CI: 1.73–19.4). The study also reported that the risk of SIL in HPV-positive women with a high-risk microbial pattern was significantly higher than in HPV-negative women with a low-risk microbial score (OR: 34.1, 95% CI: 4.95–284.5). These results were limited since risk factors were not evaluated, and the comparison groups combined different grades of disease severity [125]. Specific bacteria, such as *Gardnerella,* may be identified as biomarkers of cervical alterations to detect female patients with a high risk of developing persistent HPV infection/CIN and progression to cancer, while not all *Lactobacillus* species are uniformly protective [72]. On the other hand, in female patients with CC a prevalence of Fusobacterium species and a decreased abundance of *Lactobacillus* species were described. Both in women with CIN and in patients with CC, the VM was similar to that in female patients with BV [135]. Particularly, a Fusobacterium predominance was more commonly detected in CC patients and was shown to be correlated with a cytokine pattern of increased levels of IL-4 and TGF-b1 mRNA, suggesting a local immunosuppression state and supporting the concept of microbiota immunity [135]. BV is related to higher HPV infection rates, so an increase in the diversity of vaginal bacteria together with a decrease in *Lactobacillus* may be involved in the persistence of HPV infection [136]. Therefore, 16S-HTS and other methods, for example, molecular diagnostic tests such as direct probe assays and real-time PCR, can be used to analyze HPV+ women with a potential risk of developing cervical lesions or viral persistence [136]. These women may have an increased risk of persistence/progression of HPV-related lesions, so the results of such tests should impact diagnostic and therapeutic management. Another article by Piyathilake et al. correlated well-defined cytological groups of women with HSIL vs. LSIL in patients with high-risk HPV-positive infection [137]. In this study, the CST was not used to classify patients according to the VM structure but the Dirichlet multinomial mixture model was applied to partition the samples into four diverse metacommunities (partitions 1–4). Bacterial communities featuring principally *L. iners* and unclassified *Lactobacillus* species (partition 3) had a higher HSIL 1 level as compared to those with diverse-taxa-unclassified *Lactobacillus, L. iners, Allobaculum, Clostridiales,* and *Bifidobacteriaceae* (partition 1; OR 5 3.48, 95% CI: 1.27–9.55) [138]. 

## 5. Future Prospects

Despite the many studies conducted on the relationship between the microbiota and host immunity, the knowledge of the specific influence of HPV on gynecologic tumors remains limited [93]. Moreover, study of the microbiota and CC would need to be undertaken in large population samples in order to predict the development of precancerous lesions, since the progression of HPV infection to CC takes decades, and in many individuals never progresses to cancer at all [139]. Therefore, long-term longitudinal studies could allow the determination of early changes in the VM, that may help to evaluate the progression of precancerous lesions [93]. Furthermore, the novel advances in microbiota sequencing and sophisticated bioinformatics technologies have supported rapid progress in our understanding of the gut microbiota and the development of tumors [139,140]. Other research, using shotgun metagenomic sequencing, demonstrated that the VM community compositions and metagenomic profiles differed between patients with CC and individuals without cancer. Consequently, researchers understood that larger additional whole-genome shotgun studies are necessary to verify these relations [93]. Research into the VM can be further enhanced using metagenomic sequencing, rather than 16S (that sequences a specific 30S ribosome subunit of the human microbiota that is unique to prokaryotes and has regions that vary significantly between diverse species of bacteria and clusters the identified bacteria into operational taxonomic units) [121,122,141] or other targeted sequencing techniques, since these lack depth. Indeed, 16S amplification does not include microbes that lack a gene to match the primers (such as viruses, archaea, and eukaryotes). Since common medium- or large-scale VM analyses employed this technique, the role of non-bacterial constituents of the VM in HPV infection and disease has not yet been described. Therefore, proteomic analysis could be key to a more complete understanding of the VM and its influence on disease in the metabolic context [142,143]. Only after the pathogenic mechanisms of interaction between microbiota and HPV have been fully understood will it be possible to identify the most effective therapeutic strategy. Indeed, the execution of microbiota analysis in clinical practice in HPV-positive patients should make it possible to identify women at high risk of progression/persistence [120,121]. Consequently this “high risk” group could be candidates for cervical biopsy (for diagnostic confirmation), more restricted follow-up, and genotyping [144,145]. In addition, a more complete knowledge of the microbiota would permit the use of a targeted, personalized therapeutic approach using antibiotics and probiotics. Indeed, in women with BV, for example, there is a high diversity microbiota, and the usual cure is to prescribe antibiotics such as metronidazole and clindamycin. However, these antibiotics could prevent cervicovaginal recolonization by *Lactobacillus* species and could therefore, in turn, lead to relapse [144]. Indeed, high rates of BV relapse are already predictable after oral treatment with metronidazole [144]. Moreover, antibiotic therapy features both side effects [146,147,148] and a lack of efficacy due to resistant strains [149]. On the other hand, based on in vitro assays, *L. crispatus LbV 88, L. gasseri LbV 150N, L. jensenii LbV* [149], and *L. rhamnosus LbV96* strains, were selected as relevant for vaginal health [150]. Consequently, a pilot clinical trial was performed in which a yoghurt preparation containing those beneficial microbes was administered to BV-diagnosed patients together with metronidazole. The study revealed that the group receiving probiotics had a considerably improved recovery rate from BV as compared to women treated only with antibiotics [151]. Likewise, in the literature, administration of probiotics is also helpful in vivo and in vitro to achieve a major CIN regression and total HPV clearance. In fact, the administration of probiotics for the manipulation of the microbiota may be a feasible possibility to induce HPV infection clearance and prevent progression to CC. Indeed, the authors revealed that the group receiving long-term probiotic treatment (6 months) showed not only a significantly higher chance of resolving cytological abnormalities but also presented increased rates of HPV clearance as compared to the group receiving short-term probiotic therapy (3 months) [152,153]. Although further clinical trials are required to elucidate this relationship in patients with high-risk microbiota, the concomitant intake of antibiotics and probiotics could lead to an effective impact on the microbiota, avoiding the persistence of HPV-related lesions and progression to CC [151]. Instead, the literature on the impact of vaccination on the microbiota is still limited. Particularly, a recent small Phase I study was performed to test whether the VM could influence vaccine responses and VM composition in women with biopsy-proven high-grade squamous intraepithelial lesions. There was no difference in bacterial diversity, but a significant increase in circulating T-helper type 1 cells, and a significant decrease in the HPV 16 viral load were reported. Further studies are needed to examine the role of the VM in response to HPV therapeutic vaccines [154]. Much evidence is available related to the microbiota, but there are still many shadows that require more sophisticated tools to gain a more prolonged insight. In particular, study of the microbiota with metagenomics may be the future roadmap, but it is also important to emphasize that these techniques are expensive and require specific funds for such research. This review of the literature was written to illustrate how a better understanding of the microbiota could be the key to personalized and specific management of cervical precancers, so even if the road is long, it may be worth investing in this direction.

## Figures and Tables

**Figure 1 biology-11-01114-f001:**
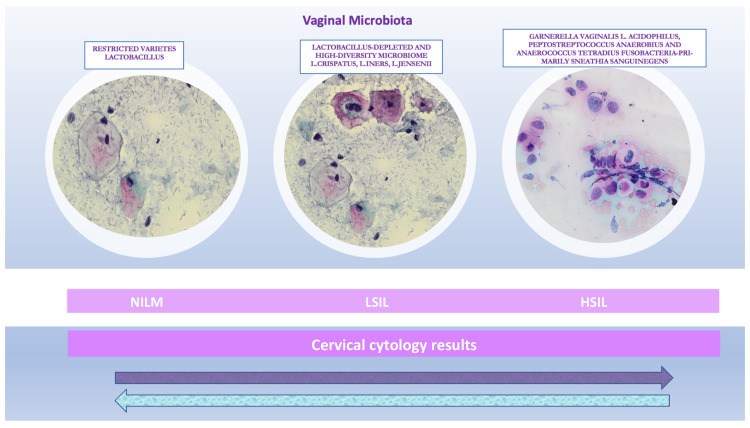
Cervical cytology results and corresponding cervical microbiota. The illustration is a summary chart of the uterine cervix microbiota community composition in physiological and precancerous stages, showing how it can influence both intraepithelial lesions progression and the stable condition of the cervical epithelium (cytological images at high magnification 40×). The arrows indicate the reversibility of the cervical changes: precancerous stages versus negative for intraepithelial lesion or malignancy (NILM).

## Data Availability

Not applicable.

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
