# Peer review of "Association between Cervical Microbiota and HPV: Could This Be the Key to Complete Cervical Cancer Eradication?"

_biology, 2022, doi:10.3390/biology11081114_

Round 1

Reviewer 1 Report

Will be updated.

Author Response

We have carried out more carefully what was suggested by the reviewer and we have more involved our gynecological pathologist (Prof. L. Resta), integrating what was requested and trying to be explicit but without going further; infact the first two paragraphs want to be introductory to the elaboration of the article and to the discussion of the chosen problem. The wording of some sentences has also been revised. We hope to have been more complete than the previous version and thank the reviewer for the valuable suggestions provided.

Reviewer 2 Report

This is a resubmission. The authors have addressed all my comments. Thank you!

Author Response

Thanks for the comment.

Round 2

Reviewer 1 Report

Please find suggestions to your revised version of the manuscript in the attachment. Thank you

Author Response

We know that it is never easy to do a literature review, so we are very grateful to the reviewer for his suggestions that have improved our manuscript to a higher level. All revisions were accepted and a major revision of chapter " Principal biomarkers for cervical cancer " was done as required. All changes are highlighted in red in the text.

This manuscript is a resubmission of an earlier submission. The following is a list of the peer review reports and author responses from that submission.

Round 1

Reviewer 1 Report

Please find comments and suggestions to the manuscript in the attachment.

Reviewer 2 Report

The concept of cervical microbiota and HPV in the development of cervical cancer or CINs is interesting.

The title of the manuscirpt that attracted me at the first glance, however, 1) the current manuscript cannot give a clear description and evidence support for the concept. 2) The epidemic data should be clearly interpretated, especially the behined rationality and population. 3) To be a review article alone, the format of this manuscript requires deliberately organized.

  1.  
  2.